# Physiotherapists’ Ethical Climate and Work Satisfaction: A STROBE-Compliant Cross-Sectional Study

**DOI:** 10.3390/healthcare11192631

**Published:** 2023-09-27

**Authors:** Amalia Sillero Sillero, Raquel Ayuso Margañon, Noemí Moreno-Segura, Juan J. Carrasco, Hady Atef, Sonia Ayuso Margañon, Elena Marques-Sule

**Affiliations:** 1Mar Nursing School (ESIMar), Parc de Salut Mar, University Pompeu Fabra Affiliated, 08003 Barcelona, Spain; asillero@esimar.edu.es; 2Social Determinants and Health Education Research Group (SDHEd), Hospital del Mar Medical Research Institute (IMIM), 08003 Barcelona, Spain; 3Department of Physiotherapy, Faculty of Physiotherapy, University of Valencia, 46010 Valencia, Spain; noemi.moreno@uv.es; 4Physiotherapy in Motion, Multispeciality Research Group (PTinMOTION), Department of Physiotherapy, University of Valencia, 46010 Valencia, Spain; juan.j.carrasco@uv.es (J.J.C.); elena.marques@uv.es (E.M.-S.); 5Intelligent Data Analysis Laboratory, University of Valencia, 46100 Valencia, Spain; 6School of Allied Health Professions, Faculty of Medicine and Health Sciences, Keele University, Staffordshire ST5 5BG, UK; hady612@cu.edu.eg; 7Department of Physical Therapy for Cardiovascular/Respiratory Disorders and Geriatrics, Faculty of Physical Therapy, Cairo University, Cairo 11432, Egypt; 8Department of Public Health Nursing, Mental Health and Perinatal Nursing, Faculty of Medicine and Health Sciences, 08907 Barcelona, Spain; soniaayusom@ub.edu

**Keywords:** physiotherapists, ethical climate, moral sensibility, job satisfaction, physical therapists

## Abstract

(1) Background: This study aimed to examine the relationship between Spanish physical therapists’ perceptions of the ethical climate, their moral sensitivity (awareness of ethical issues), and job satisfaction. (2) Methods: the study analyzed descriptive correlational data on 104 physical therapists from three Spanish metropolitan hospitals. Respondents completed a demographic data form, an ethical climate questionnaire, a job satisfaction survey, and a moral sensitivity scale. This study complies with the Strengthening the Reporting of Observational Studies in Epidemiology (STROBE) guidelines. (3) Results: With a mean score of 4.2, physical therapists are typically content with their jobs. The mean scores for the moral sensitivity and ethical climate questionnaires are high, at 40.1 (SD 6.3) and 96.8 (SD 17.1), respectively. There is a significant positive correlation between job satisfaction and ethical climate (r between 0.59 and 0.79) but only a weak correlation between job satisfaction and moral sensibility (r between 0 and 0.32 for the three aspects measured). (4) Conclusions: Generally speaking, physical therapists reported that they had high job satisfaction, a positive workplace environment, and excellent management support. Despite a weak relationship with moral sensibility, there is a strong association between ethical behavior, hospital organization, and higher levels of job satisfaction. It is important to encourage the development of moral sensibilities to boost psychological well-being and therapeutic decision-making.

## 1. Introduction

Physical therapists often encounter ethical concerns in their practice due to the complex nature of treatment and their moral responsibilities [1]. Some documented ethical problems include the lack of resources, discrimination based on personal characteristics, and unethical behavior displayed by other health professionals [2,3]. Furthermore, ethical dilemmas arise concerning delivering high-quality care, adherence to patients’ rights and organizational expectations, and promoting patient autonomy. This can influence employees’ job satisfaction, understood as the emotion experienced, and affect their productivity. The COVID-19 pandemic has further heightened some of these challenges. Given the numerous ethical issues faced by physiotherapists in clinical practice, it is crucial to provide them with adequate support to navigate the complexity, ambiguity, and occasional messiness encountered in their daily work. Without such support, they may experience moral distress and burnout and even consider leaving the profession [4,5,6]. Consequently, physical therapists must effectively develop strategies to address these ethical tensions and challenges.

When making decisions in their clinical work, physical therapists refer to the corresponding codes of professional ethics, with the ethical guidelines provided by the International Physiotherapy Confederation serving as the foundation. However, ethical decisions in practice can be influenced by various factors, including the ethical climate of the healthcare organization and the moral sensitivity of the therapists [7]. Consequently, the environments and demands of the workplaces where physiotherapists operate can make it impossible to comply with codes of ethics [8]. These factors can ultimately impact job satisfaction and significantly affect workplace retention.

The ethical climate in healthcare settings has been examined from various philosophical perspectives. Victor and Cullen (1988) defined it as the employees’ understanding of what is ethically correct in their interventions and how ethical issues are addressed within the organization. This concept includes the procedures and policies that shape expectations for ethical behavior [9].

Olson (1995) described the ethical climate as a flexible and adaptable organizational factor that can enhance the work environment, highlighting its importance in healthcare settings [10]. According to Olson, the ethical climate is assessed through interprofessional relationships and the ethical decision-making abilities of individuals involved, including peers, patients, hospitals, managers, and physicians. Essentially, the ethical climate refers to individuals’ perceptions of how ethical issues are handled in their work environment [11].

A supportive ethical climate fosters the sharing of experiences and collaboration among healthcare professionals, enabling them to address patients’ needs effectively. This type of environment encourages ethical dilemmas to be deliberated upon using ethical principles, facilitating the process of ethical decision-making. Conversely, a hostile ethical climate contributes to increased moral stress and obstructs ethical deliberation. Improving the ethical climate in healthcare organizations is crucial for delivering superior medical care and promoting job satisfaction among healthcare professionals [12]. A positive ethical climate can contribute to a more positive work environment, better patient care, and higher levels of job satisfaction for healthcare workers.

Researchers worldwide have shown a growing interest in the ethical climate and its correlation with various variables, including job satisfaction. While the importance of the hospital’s ethical climate and its impact on job satisfaction among healthcare professionals has been demonstrated, this area has yet to be explored.

Moral awareness and sensitivity are essential for making ethical decisions in healthcare. Moral awareness involves understanding how one’s perspectives and actions affect their surroundings, especially about ethical issues. On the other hand, moral sensitivity, as defined by Lützen, refers to the ability to recognize moral dilemmas and comprehend the consequences of decisions made on behalf of the patient [13]. It serves as a foundation for exhibiting moral behavior that enhances the standard of care [14].

Creating a favorable ethical climate in healthcare organizations is crucial for enabling professionals to effectively address ethical issues. An ethical climate that supports moral sensitivity and ethical decision making can positively impact the quality of care provided and the job satisfaction of healthcare professionals [15].

While there is a significant body of research on moral sensitivity in patient care, job satisfaction, and perceptions of the ethical environment in nursing within the hospital context [16,17], there has been relatively limited research on this topic, specifically among physiotherapists. Thus, the study mentioned aimed to fill this gap by examining the association between job satisfaction, moral sensitivity, and the ethical climate in hospital physiotherapy using Linda Orson’s Hospital Ethical Climate Survey (HECS) instrument.

By investigating this relationship, we can better understand how the ethical climate and moral sensitivity influence job satisfaction among physical therapists, ultimately providing valuable insights for healthcare organizations and professionals in improving the work environment and patient care quality. Therefore, this study aimed to examine the relationship between the perception of the ethical climate, moral sensitivity, and job satisfaction concerning the practice of physical therapists working in a hospital setting.

## 2. Materials and Methods

### 2.1. Research Design and Setting

An analytic transversal observational design was used to conduct this study at the University of Valencia, Valencia (Spain). All participants were informed about the purpose and procedures and provided written informed consent before participating. This study complied with the Strengthening the Reporting of Observational Studies in Epidemiology (STROBE) guidelines [18].

### 2.2. Sample

Sample size calculations were performed using G* Power software (version 3.1.9.7; Heinrich-Heine-Universität, Düsseldorf, Germany). For a bivariate correlation, assuming a minimum correlation ρ(H1) = 0.3, α = 0.05, and β = 0.20, a total sample of 84 subjects was needed. Furthermore, for linear multiple regression, assuming a medium effect size f^2^ = 0.15, α = 0.05, β = 0.20, and 8 predictors, a sample size of 109 participants was required.

Finally, a convenience sample of 104 physical therapists working at 3 different metropolitan hospitals in Valencia was recruited. The inclusion criterion was working as a physical therapist in a hospital setting with one year of experience or more, and they were excluded if they were unemployed. We implemented a criterion for minimum working time to ensure a diverse representation of experience levels within physiotherapy. This deliberate criterion was designed to encompass a spectrum of experience, recognizing that a physiotherapist with 1 year of experience might have different insights and perspectives compared to someone with 33 years of experience.

### 2.3. Procedure

The study was conducted from October 2021 to April 2022. All the participants were evaluated in one face-to-face evaluation session, where all the participants filled out printed self-administered questionnaires. Each physical therapist took about 60 min to complete the questionnaires.

### 2.4. Outcomes

Sociodemographic data such as age, gender, civil status, educational level, ethical course participation, work experience, and type of contract were collected. Moreover, the following variables were measured.

Ethical climate was evaluated through the Hospital Ethical Climate Survey (HECS), developed by Olson (1995) in the USA. This is a 26-item self-administered questionnaire where the responder answers on a 5-point Likert scale from 1, “almost never true”, to 5, “almost always true”. It consists of 5 subscales that examine relations between the respondent and their peers (for example, listening to each other, helping each other, their colleagues’ competence in care, and whether they provide safe patient care), patients (recognizing the expectations and wishes of the patients), managers (support and help received from managers), hospital (supportive hospital policies), and physicians (related to the respect of physical therapists and physicians for each other and each other’s decisions, and the physicians’ asking for physical therapists’ opinions in decisions about patient care). The total score obtained ranged from 26 to 130 points [10]. The higher scores imply that the hospital’s ethical climate perception is high. The Cronbach’s alpha internal consistency and reliability coefficient for the HESC scale was 0.91.

Job satisfaction was assessed through the 1-item Overall Job Satisfaction survey, in which participants were asked to rate their level of overall job satisfaction on a scale of 1 to 7, where a score of 1 implies being extremely dissatisfied and 7 implies being extremely satisfied. This scale shows good levels of reliability (r = 0.73) and a correlation of 0.82 with 15-item Job Satisfaction [19].

Moral sensitivity was measured with the Revised Moral Sensitivity Questionnaire (RMSQ), which includes 9 items using a 6-point Likert scale, where 1 implies totally disagree and 6 totally agree. The RMSQ comprises 3 dimensions: a sense of moral burden, moral strength, and moral responsibility [20]. Total scores range from 9 points (lower moral sensitivity) to 54 points (higher moral sensitivity). The questionnaire has been proven to be a valid and reliable instrument (α = 0.83) [19] to assess moral sensitivity [13].

### 2.5. Data Analysis

Data were coded by the researcher and statistically analyzed using SPSS (Statistical Package for the Social Science) version 26 (SPSS Inc., Chicago, IL, USA). The Kolmogorov–Smirnov test was used to verify the normality of the continuous data. Descriptive results are presented as mean, standard deviation, median, 25 and 75th percentile, minimum, maximum, and frequencies. The association between job satisfaction, ethical climate (total score as well as peers, patients, managers, hospital, and physicians’ dimensions), and moral sensitivity (total score as well as moral burden, moral strength, and moral responsibility dimensions) questionnaires were assessed using Pearson’s or Spearman’s correlation depending on the distribution of the data. Correlation coefficients were interpreted as small (0.1–0.3), medium (0.3–0.5), and large (0.5–1.0). Finally, a stepwise multiple linear regression analysis was performed to investigate the predictive power of independent variables (age, gender, ethical climate, and moral sensitivity dimensions) on job satisfaction (dependent variable). Statistical significance was set at *p* < 0.05.

### 2.6. Ethical Considerations

Approval was obtained from the Institutional Review Board of the University of Valencia, Spain (IE1544051), and all procedures were conducted according to the principles of the Declaration of Helsinki (October 2013, Fortaleza, Brazil). The researchers explained the aim of the research to all participants [21]. The privacy and confidentiality of data were maintained. The anonymity of participants was granted. Informed consent to participate was collected before data collection.

## 3. Results

A total of 104 physical therapists (45 men and 59 women) completed all the questionnaires. The mean (SD) age was 35.0 (9.5) years old, with a mean of 11.7 (8.2) years of work experience. Notably, 93% of the participants reported not having taken an ethics course. The physical therapists are generally satisfied at work, with a mean score of 4.2 (1.0). The total scores in the ethical climate and moral sensitivity questionnaire are high, with a mean of 96.8 (17.1) and 40.1 (6.3), respectively. Table 1 shows the complete results of the demographic data and questionnaire dimensions.

The results of the correlation analysis are depicted in Figure 1. The correlation between pairs of variables is shown in the rectangles at the intersections between rows and columns. Work satisfaction (column three) strongly correlates with the ethical climate dimensions and the total score (rows four to nine, r between 0.59 and 0.79). However, work satisfaction only weakly correlates with the moral sensibility strength and responsibility dimensions (rows ten to thirteen, r = 0.32 and r = 0.26, respectively). Furthermore, correlations between dimensions and the total score within each questionnaire are strong. For the ethical climate, managers (column six, r = 0.88), and hospital (column seven, r = 0.88) dimensions are the most correlated with the ethical climate total score (row nine). Regarding moral sensibility, the most correlated dimension with the total score (row thirteen) is a moral burden (row ten, r = 0.85). Moreover, the moral burden does not correlate significatively with ethical climate dimensions (columns four to nine). Moral strength (row eleven) presents a moderately significant correlation with ethical climate dimensions (r between 0.32 and 0.43, except for physicians, for which it is not significant.). Moral responsibility (row eleven) only shows a weak correlation with ethical patients (r = 0.26) and total ethical climate (r = 0.20). Finally, the total moral sensibility score only presents a weak correlation with ethical peers (r = 0.21). Lastly, according to the post hoc power analysis, with 104 subjects, significant correlations greater than r = 0.27 have a statistical power greater than 0.80.

The stepwise regression models obtained are shown in Table 2. The best model includes the total ethics score and age as independent variables, showing a significant (*p* < 0.05) association with job satisfaction (adjusted R^2^ = 0.67). According to the post hoc power analysis, with an effect size = 2.03 and 104 participants, the statistical power is 1.

## 4. Discussion

The present study investigated the relationship between the perception of the ethical climate, moral sensitivity, and job satisfaction among physical therapists in a hospital setting. According to our knowledge, this study is the first to examine the association between physical therapists’ perceptions of the ethical climate, moral sensitivity, and job satisfaction.

The findings of our study indicate a positive work ethic climate among physical therapists. However, it is essential to recognize that individuals’ perceptions of their organization’s procedures and functions can vary, even within ostensibly similar cultural and organizational contexts [22,23,24]. Therefore, it becomes important to concentrate on fostering the ideal environment for addressing the ethical climate’s key components within the context of the hospital to ensure quality care.

The ethical climate in our study was generally rated higher than the managers’ and hospitals’ ethical climates [25]. We observed that physiotherapists felt supported and valued by the organization, indicating they shared their objective of implementing the necessary management procedures and receiving assistance in conflict resolutions. They also felt that their judgments were taken into consideration.

Similar results have been found in nursing, where hospital regulations are crucial for a supportive ethical environment [26]. Our study also highlighted certain elements, such as productivity standards, billing practices, and organizational aspects of the hospital, as areas of concern for physical therapists, leading the perception of being financially driven at the expense of patient care [22]. Also, they did not feel at ease with the creation of some organizational policies according to the practical ethical frameworks they observed throughout the pandemic [27].

The lowest scores corresponded to the dimensions of peers and doctors; this result is probably related to the power relations developed between the different professional ranges [28]. Previous research has confirmed that when contributions are inferior to those of doctors, it negatively affects relationships among colleagues and hampers collaboration in patient care [29]. To attain shared goals, actions must be taken to enhance labor relations because doing so can interfere with keeping a favorable ethical climate and, consequently, with the standard of care.

An interesting observation from our study is that most participants reported not having undergone formal ethics education. However, it is noteworthy that their perceptions of the ethical climate have not been negatively influenced. This finding suggests that other factors might play a role in developing ethical awareness and decision making among physical therapists beyond formal ethics education [30].

On the positive side, our study reported high average scores for job satisfaction among physical therapists at the hospital, in line with other studies [31,32]. Notably, we observed strong correlations between all aspects of the ethical climate and job satisfaction, with a surprising finding that the ethical climate with coworkers and physicians was less significant in determining job satisfaction. This contrasts with earlier research suggesting that professionals who receive peer support and experience team cohesion exhibit higher job satisfaction [33]. In Kota’s examination of the variables influencing physical therapists’ job satisfaction, Kota emphasized the importance of interpersonal relationships [34], which may go beyond the belief held by physical therapists in our study regarding these dimensions and a less favorable ethical climate.

Furthermore, the relationship with managers is significant because it helps physical therapists feel more satisfied with their jobs. Other studies have identified a climate of justice among peers [35] and leaders’ positive and consistent feedback to employees [36] as factors that boost job satisfaction. Regarding the patient dimension, other studies revealed that professionals were very satisfied when evidence-based practices were used [37], when the professional’s dedication to the patient was shown [38], or when they thought they were providing a level of care that was in line with patient expectations [39].

Regarding hospital dimensions, support programs to prevent burnout or to improve work environments and employee well-being and the ability to decide on patient care and workforce problems are elements previously reported as determinants of professional job satisfaction [40,41]. Therefore, the evidence supports the importance of focusing on the job satisfaction of physical therapists, considering the constitutive elements of a positive ethical climate.

Among the demographic variables analyzed, age had a higher significance in the regression model as a predictor of job satisfaction. Age would explain moderate to high levels (67%) of job satisfaction in hospital physical therapists, with older professionals being the most satisfied, which reveals a pattern consistent with other studies in the context of healthcare work [42]. This could be because older professionals have more experience. It allows them to better adapt to the job and obtain a more objective view of what causes dissatisfaction among other colleagues, for example, the relationship between them or with the doctors or hospital managers and organizational issues [43]. In addition to relationships, other factors such as recognition and salary have been evidenced in the higher job satisfaction of health workers, including physical therapists [33,34,35,36,37,38,39,40,41,42,43,44].

Compared to younger physical therapists, the older ones may have reached their salary targets and professional recognition, resulting in higher satisfaction. The ethical climate also showed a significant association (*p* < 0.05) with job satisfaction (adjusted R^2^ = 0.67). Each point in the total ethical climate increased job satisfaction by 0.049 points, and each year of age increased satisfaction by 0.016 points; therefore, improving the ethical climate regarding relationships and improving recognition and the salary aspect seems to be necessary to increase the satisfaction of hospital physical therapists in the organization.

Regarding moral sensitivity, the dimension of moral burden was the one that obtained the highest score (mean (SD) = 16.1 (3.4), r = 0.85), understood as the perceived burden of hospital physical therapists given the difficulty of distinguishing the needs of the patient, challenges in managing feelings in the face of the suffering of the patient and maintaining the balance between the potential to do good and the risk of causing harm [13].

A moral burden is accompanied by anguish or moral suffering that can interfere with the quality of care [45]. Prior research has shown an inverse relationship between moral suffering and the ethical climate, implying that a more positive ethical climate results in lower moral suffering and increased job satisfaction [22,46]. Thus, it is important to develop policies that promote a favorable ethical climate among hospital physical therapists.

While using a single-item questionnaire for job satisfaction may have limitations in capturing the full complexity of this construct, our findings still highlight the significance of the ethical climate in promoting positive job experiences for physical therapists. This implies that even with the single-item measurement, we could identify meaningful associations between job satisfaction and ethical climate dimensions. However, a more comprehensive and multidimensional approach to assessing job satisfaction in future research could provide a more in-depth understanding of its relationships with the ethical climate and other contributing factors among healthcare professionals.

### 4.1. Limitations and Strengths

However, the following limitations should be considered. Convenience sampling was used initially, which has little generalizability. However, due to the constrained number of physical therapists employed in each hospital, our data were gathered from different hospitals in Valencia, Spain.

Second, non-response bias and response bias, common drawbacks of self-report research questionnaires, are present. Since everyone who participated in the study responded, this might not be a serious issue. However, some participants may have thought about not responding to the questionnaire so as not to be criticized by their institution. Yet, this is mitigated by the respondents’ and the replies’ confidentiality and anonymity.

Furthermore, the variables investigated in the study are not meant to be exhaustive or complete. Since the single-question questionnaire might not fully reflect the variability in participants’ job satisfaction levels, it can be stressed that results related to job satisfaction should be interpreted with caution. For example, job satisfaction is only one of several possible organizational outcomes. Other variables that future research can focus on include staff motivation and commitment. Other possibilities include the existence or implementation of some measures of staff attributes and ethical attitudes. However, it is noted that a lengthy research questionnaire is likely to make the questionnaire more challenging to complete and hence is expected to reduce the response rate.

The main strength of our manuscript lies in being the first to investigate the relationship between physiotherapists’ perceptions of ethical climate, moral sensitivity, and job satisfaction. Prior to our study, this specific aspect had not been explored, and our research fills a significant gap in the existing literature. This understanding is essential for healthcare organizations and professionals to create a more nurturing and supportive work environment, which ultimately leads to better patient care.

### 4.2. Implications of Findings

This study provides potentially useful notes on how management might raise job satisfaction by enhancing the ethical climate within the workplace. The study might encourage management professionals to adopt actions that enhance physical therapists’ moral conduct.

The findings might encourage management professionals to adopt actions that raise workers’ ethical standards. This study shows how the ethical atmosphere, job satisfaction, and moral sensibility are related. Because this topic has not been investigated in other countries, it should help advance future research in this area, especially in the population of physical therapists. Among the elements on which the organization should focus its efforts are the relationship between workers and managers, offering supportive policies, and favoring the relationship between colleagues. They should also pay attention to younger physical therapists, improving salaries and relationships. Finally, policies should be developed that promote an ethical climate that alleviates the moral burden of patient care. Hence, the current study might be of interest to scholars, managers, and practitioners in this field.

## 5. Conclusions

This study has shed light on the relationship between the perception of the ethical climate, moral sensitivity, and job satisfaction among physical therapists in a hospital setting. The findings revealed a positive ethical work climate among physiotherapists, consistent with similar studies in the field. It underscores the significance of fostering a supportive and nurturing work environment within hospitals to ensure quality care and job satisfaction.

Furthermore, the physical therapists in our study reported satisfaction with their work, expressing that the hospital’s work climate was favorable and that they perceived strong management support. These factors contributed to a strong association between ethical behavior, the hospital organization (managers and hospital), and higher levels of job satisfaction, despite a weaker association with moral sensibility.

Although the impact of moral sensitivity on ethical climate and job satisfaction was not as pronounced in our study, it indicates the need for further research to gain a more profound understanding of its role in influencing job satisfaction and ethical decision-making.

We recommend hospital management continue implementing policies for a positive ethical climate in the work environment so that physical therapists can maintain their continuity and satisfaction with the work. Also, it is necessary to promote the formation of higher moral sensibility, which could increase psychosocial well-being and improve decision-making in clinical practice.

## Figures and Tables

**Figure 1 healthcare-11-02631-f001:**
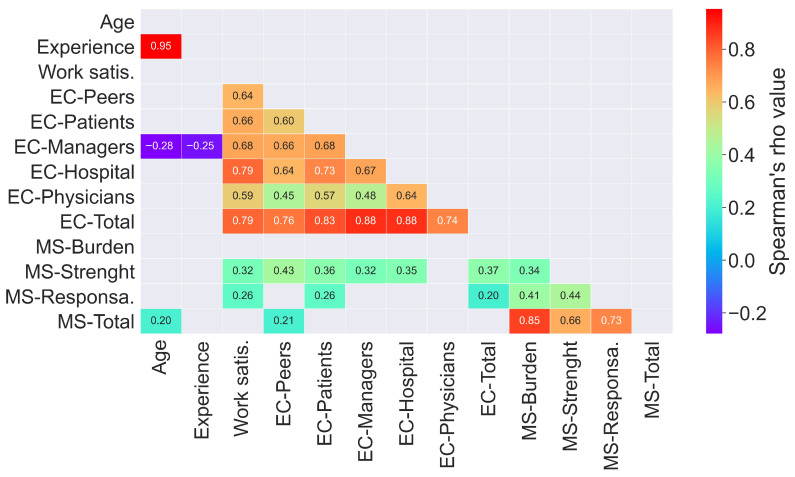
Heatmap of correlation analysis. Experience: years of experience as a physiotherapist; Work satis.: work satisfaction; EC: ethical climate; MS: moral sensibility; Responsa: responsibility; Only significant correlation values with a *p* < 0.05 are shown.

**Table 1 healthcare-11-02631-t001:** Demographics, descriptive data, and questionnaire results.

		Mean	SD	Median	25th	75th	Min	Max
	Age (years)	35.0	9.5	34.0	26.0	40.0	22	62
	Experience (years)	11.7	8.2	12.0	3.3	18.0	1	33
	Gender, n (%)	Men: 45 (43.3), women: 59 (56.7)						
	Civil status, n (%)	Married: 36 (34.6), single: 51 (49.0), common-low marriage: 13 (12.5), divorced/separated: 4 (3.8)						
	Education level, n (%)	Diploma: 33 (31.7), Graduate: 11 (10.6), Postgraduate: 16 (15.4), Master: 35 (33.7), Ph.D.: 9 (8.7)						
	Ethics course, n (%)	No: 93 (89.4), yes: 11 (10.6)						
	Contract type, n (%)	Owner: 30 (28.8), temporary: 73 (70.2), other: 1 (1.0)						
	Overall job satisfaction	4.2	1.0	4.0	4.0	5.0	2	6
Hospital Ethical Climate Survey	Peers	17.0	2.9	18.0	15.3	19.0	7	20
	Patients	15.5	2.5	16.0	14.0	17.8	7	20
	Managers	23.0	5.8	24.0	18.0	28.0	9	30
	Hospital	22.2	4.5	23.0	20.0	25.0	7	30
	Physicians	19.2	4.3	20.0	16.3	22.0	7	30
	Total	96.8	17.1	100.0	85.5	109.0	39	130
Moral Sensitivity Questionnaire	Burden	16.1	3.4	17.0	14.0	18.0	6	23
	Strength	14.8	2.4	15.0	14.0	16.0	4	18
	Responsibility	9.2	1.9	9.0	8.0	11.0	4	12
	Total	40.1	6.3	41.0	37.3	44.0	14	51

SD: Standard Deviation; 25th: 25th percentile; 75th: 75th percentile; Min: Minimum Value; Max: Maximum Value; n (%): number of observations in a sample (percentage).

**Table 2 healthcare-11-02631-t002:** Stepwise regression model for work satisfaction.

Model		B	SD	95.0% CI for B		β	Adjusted R^2^	SEE	*p*	f^2^
1	Constant	−0.34	0.34	−1.01	0.32				0.31	
	Ethics Total	0.05	0.01	0.04	0.05	0.81	**0.65**	0.59	**<0.001**	1.86
2	Constant	−1.04	0.42	−1.87	−0.22				0.014	
	Ethics Total	0.05	0.01	0.04	0.06	0.83			<0.001	
	Age	0.02	0.01	0.01	0.03	0.15	**0.67**	0.57	**0.008**	2.03

B: Regression coefficients and standard errors. CI: Confidence intervals. β: Standardized regression coefficients. R^2^: Coefficient of determination. SEE: Standard error of the estimate. *p*: *p*-value. f^2^: Cohen’s f-square effect size. Adjusted R^2^ and significant *p*-values are highlighted in bold. For the work satisfaction model, age, gender, ethical climate, and moral sensitivity dimensions were included as independent variables.

## Data Availability

The datasets generated and/or analyzed during the current study are not publicly available but are available from the corresponding author upon reasonable request.

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
