# Peer review of "Physiotherapists’ Ethical Climate and Work Satisfaction: A STROBE-Compliant Cross-Sectional Study"

_healthcare, 2023, doi:10.3390/healthcare11192631_

Round 1

Reviewer 1 Report

Authors developed an insteresting research about Physiotherapists’ job satisfaction and its association with other components such as ethical climate or sensitivity. 

In general, this manuscript is well written and has an introduction that provides enough background to understand their objectives; moreover, it is supported by relevant references. 

The objectives are not really clear though, so I would reccomend the authors to state them at the end of the introduction, so conclusions can be better understood. 

The methods of this manuscript are also very clear. I have some concerns about taking a "convenience sample" (line 103, page 3) but later on, authors discuss this and recognize this may be a serious limitation of their work.

I have some suggestions to modify this manuscript on the results section. In my opinion, it is difficult to see the results and to understand what the tables mean, mainly in table 2 where all the correlations are exposed. I reccomend the authors to modify these tables (1 and 2) so the reader can easier understand what's happening between outcomes and how they correlate. 

Then, the discussion is really clear, although it is a bit disorganized. When you state your objectives on the introduction, then you can organize your discussion in the same order, so it is easier for the reader to understand what you are exposing. 

Finally in the conclusions, I would suggest to change them according to the objectives, once you state them clearly in the introduction.  

Author Response

Consulte el archivo adjunto

Reviewer 2 Report

Dear authors,

At the end of the article, you conclude that physiotherapists reported being satisfied with their work and that it is associated with an ethical climate, but you fail to justify how this issue is relevant to clinical practice. I could not see the relevance of the topic, although the proposal is interesting.

I believe that both the introduction and discussion need to be redone to provide a good justification for the study and its relevance.

Author Response

Consulte el archivo adjunto

Reviewer 3 Report

Thank you for the opportunity to edit and develop your manuscript.

The study is interesting as well-being at work, in relation to ethics and morals, and how to get employees to stay at the workplace are very current topics.

However, there are some points to improve readability and understanding of the manuscript:

1. The study has an too long and difficult title. See if it can be shortened and become more interesting

2. Abstract (line 22). The aim does not match the study plan, as you have not studied job satisfaction in relation to physical therapists' professional practice

2. The introduction section lacks a clear presentation of the main concept of job satisfaction and its relevance in relation to the aspects you are studying (line 51) and then there should be a clear rationale followed by the purpose of the study

3. The discussion section lacks a discussion of the study's results regarding the 1-item questionnaire for job satisfaction (ordinal scale rating 1-7). How sensitive to change is this? does it affect the interpretation of the result?

4. There is also a need extended and clearer implications for clinical practice. What parameters do the organisations have to work with, to make physiotherapists in hospitals to feel improved job satisfaction in relation to ethics and morals

Author Response

Consulte el archivo adjunto

Reviewer 4 Report

Introduction: The introduction would benefit from highlighting the need for this type of study and how it fills the literature gap.

The primary concern of the introduction is the lack of scope for conducting this study. For example, the authors start the introduction by stating “ethical issues” faced by the physical therapist. However, they failed to provide more details about these “ethical issues” and had never mentioned if these issues are faced by them in a specific country or is a concern globally.

On the other hand, the introduction presents the Olson research; however, it is not apparent why they are focusing on this study.

Furthermore, the introduction should finish with the aims of the study. However, the aims of this study are not well articulated (Lines 85-94). The authors should use specific terminology for this type of research to attract the reader's attention and explain why this study will be conducted.

Another major flaw of this manuscript is how the authors use the present and past tense interchangeably. It also uses the pronoun “we” with “it was mentioned, " making it difficult to read.

Methods: There are significant flaws in this section.

Study design: The authors failed to describe the study design.

Sample: The significant flaw of this study is the lack of sample size and the power of the study. How was it calculated? How many participants were recruited?

Outcomes: Why were these outcomes measured? What is the significance of these outcomes?

Results: Did the study have enough power to be conducted?

The authors stated, “93% of participants have not taken an ethics course.”  First of all, a sentence cannot start with a number. Secondly, what is the relevance of taking an ethics course to this survey? The authors failed to connect the results with other research relevant to their study.

Discussion: The discussion section failed to address the interpretation of results. The discussion section was weak and did not properly support the evidence needed in the results section. There is much irrelevant literature in the discussion section that is off-topic from this study. For example, results found in two other studies were brought up in the discussion. This information is more suitable for the introduction to justify further what the results of this study will add to the literature. The limitations were provided; however, the authors failed to present the strengths of this study.

Conclusion: The authors failed to provide the power of the study, a p-value, confidence interval for the statistical analysis. Consequently, the authors should not generalize the conclusion. This is especially important considering the lack of information about the sample size, the power of the study, the physical therapists who answered the survey, and how survey results were analyzed. Additional information about services is needed to make it effectively implemented. Also, this study failed to address the impact of COVID-19 on these participants. 

A supportive ethical climate promotes sharing experiences and collaboration among healthcare professionals to meet the needs of patients. It creates an environment where ethical dilemmas are deliberated based on principles and ethical decision-making is facilitated. On the other hand, a hostile ethical climate increases moral stress and hampers the ethical deliberation process. Improving the ethical climate in healthcare organizations is crucial for providing better medical attention and promoting job satisfaction.

Author Response

Consulte el archivo adjunto

Reviewer 5 Report

Dear Authors,

It is my pleasure to review your study but I have a few doubts.

Generał information:

-in the first place, the article should be prepared in accordance with the guidelines of the journal, the references should be corrected

Introduction:

-the purpose of the study should be presented clearly and transparently. It should be corrected. 

M&M:

-the inclusion and exclusion criteria should be clarified - it should be corrected. Has the minimum working time in the profession of a physiotherapist been adopted? This plays a very important role. A physiotherapist working for a year has a different experience than 33 years. This should be explained.

-was the sample size calculated?

Results:

-abbreviations in all tables should be explained,

Discussion:

-in line 217: "It emphasized a positive work ethic climate, which is consistent with findings from other references that were also regarded favorably [21,22]."

I don't agree with that. 

In reference 22, for example, there is a different range of age and work experience, which shows more credibility than in your work. I think that should be changed. The cited literature does NOT confirm your research.

Conclusions:

-Conclusions should be more correlated with the aims of the study.

References:

-no DOI in references, it should be corrected.

I have serious doubts about the reliability of the results. 1. Too large age range of the respondents. Experience in the range of 1-33 years can significantly affect the results.

I propose to increase the number of respondents and improve the inclusion criteria.

English proofreading recommended.

Author Response

Consulte el archivo adjunto

Round 2

Reviewer 4 Report

The authors addressed all my suggestions.

Author Response

Respuesta del revisor 4 comentarios: 

Agradecemos los comentarios del revisor y nos alegra haber resuelto las sugerencias de mejora .

Reviewer 5 Report

Dear Authors,

The manuscript was only partially corrected.

References have not been corrected in accordance with journal guidelines. 

The aim of the study should be at the end of the introduction.

The inclusion criteria still require correction. 

I believe that the age range 1-33 can have a significant impact on the results. The minimum age is 22 and maximum 62 makes a significant difference in experience. The research methodology is not correct. Especially that the study is related to work experience.

I propose to modify this, reduce the scope of the experiment and resubmit the results.

I propose to modify this and re-present the results.

Author Response

Consulte el archivo adjunto
